# Adversarial Reprogramming of Neural Networks

**Gamaleldin F. Elsayed**[*]
Google Brain
gamaleldin.elsayed@gmail.com

**Ian Goodfellow**
Google Brain
goodfellow@google.com

**Jascha Sohl-Dickstein**
Google Brain
jaschasd@google.com

## Abstract

Deep neural networks are susceptible to *adversarial* attacks. In computer vision, well-crafted perturbations to images can cause neural networks to make mistakes such as confusing a cat with a computer. Previous adversarial attacks have been designed to degrade performance of models or cause machine learning models to produce specific outputs chosen ahead of time by the attacker. We introduce attacks that instead *reprogram* the target model to perform a task chosen by the attacker—without the attacker needing to specify or compute the desired output for each test-time input. This attack finds a single adversarial perturbation, that can be added to all test-time inputs to a machine learning model in order to cause the model to perform a task chosen by the adversary—even if the model was not trained to do this task. These perturbations can thus be considered a program for the new task. We demonstrate adversarial reprogramming on six ImageNet classification models, repurposing these models to perform a counting task, as well as classification tasks: classification of MNIST and CIFAR-10 examples presented as inputs to the ImageNet model.

## 1 Introduction

The study of adversarial examples is often motivated in terms of the danger posed by an attacker whose goal is to cause model prediction errors with a small change to the model's input. Such an attacker could make a self-driving car react to a phantom stop sign (Evtimov et al., 2017) by means of a sticker (a small $L_0$ perturbation), or cause an insurance company's damage model to overestimate the claim value from the resulting accident by subtly doctoring photos of the damage (a small $L_\infty$ perturbation). With this context, various methods have been proposed both to construct (Szegedy et al., 2013; Papernot et al., 2015; 2017; 2016; Brown et al., 2017; Liu et al., 2016) and defend against (Goodfellow et al., 2014; Kurakin et al., 2016; Madry et al., 2017; Tramèr et al., 2017; Kolter & Wong, 2017; Kannan et al., 2018) this style of adversarial attack. Thus far, the majority of adversarial attacks have consisted of *untargeted* attacks that aim to degrade the performance of a model without necessarily requiring it to produce a specific output, or *targeted* attacks in which the attacker designs an adversarial perturbation to produce a specific output for that input. For example, an attack against a classifier might target a specific desired output class for each input image, or an attack against a reinforcement learning agent might induce that agent to enter a specific state (Lin et al., 2017).

In practice, there is no requirement that adversarial attacks will adhere to this framework. Thus, it is crucial to proactively anticipate other unexplored adversarial goals in order to make machine learning systems more secure. In this work, we consider a novel and more challenging adversarial goal: reprogramming the model to perform a task chosen by the attacker, without the attacker needing to compute the specific desired output. Consider a model trained to perform some *original task*: for inputs $x$ it produces outputs $f(x)$. Consider an adversary who wishes to perform an *adversarial task*:

---

[*]Work done as a member of the Google AI Residency program (g.co/airesidency).

for inputs $\tilde{x}$ (not necessarily in the same domain as $x$) the adversary wishes to compute a function $g(\tilde{x})$. We show that an adversary can accomplish this by learning *adversarial reprogramming functions* $h_f(\cdot; \theta)$ and $h_g(\cdot; \theta)$ that map between the two tasks. Here, $h_f$ converts inputs from the domain of $\tilde{x}$ into the domain of $x$ (i.e., $h_f(\tilde{x}; \theta)$ is a valid input to the function $f$), while $h_g$ maps output of $f(h(\tilde{x}; \theta))$ back to outputs of $g(\tilde{x})$. The parameters $\theta$ of the adversarial program are then adjusted to achieve $h_g(f(h_f(\tilde{x}))) = g(\tilde{x})$.

In our work, for simplicity, we define $\tilde{x}$ to be a small image, $g$ a function that processes small images, $x$ a large image, and $f$ a function that processes large images. Our function $h_f$ then just consists of drawing $x$ in the center of the large image and $\theta$ in the borders (though see Section 4.5 for other schemes), and $h_g$ is simply a hard coded mapping between output class labels. However, the idea is more general; $h_f$ ($h_g$) could be any consistent transformation that converts between the input (output) formats for the two tasks and causes the model to perform the adversarial task.

We refer to the class of attacks where a model is repurposed to perform a new task as *adversarial reprogramming*. We refer to $\theta$ as an *adversarial program*. In contrast to most previous adversarial work, the attack does not need to be imperceptible to humans, or even subtle, in order to be considered a success. However, we note that it is still possible to construct reprogramming attacks that are imperceptible. Potential consequences of adversarial reprogramming include theft of computational resources from public facing services, repurposing of AI-driven assistants into spies or spam bots, and abusing machine learning services for tasks violating the ethical principles of system providers. Risks stemming from this type of attack are discussed in more detail in Section 5.2.

It may seem unlikely that an additive offset to a neural network's input would be sufficient on its own to repurpose the network to a new task. However, this flexibility stemming only from changes to a network's inputs is consistent with results on the expressive power of deep neural networks. For instance, in Raghu et al. (2016) it is shown that, depending on network hyperparameters, the number of unique output patterns achievable by moving along a one-dimensional trajectory in input space increases exponentially with network depth. Further, Li et al. (2018) shows that networks can often be trained to high accuracy even if parameter updates are restricted to occur only in a low dimensional subspace. An additive offset to a neural network's input is equivalent to a modification of its first layer biases (for a convolutional network with biases shared across space, this operation effectively introduces new parameters because the additive input is not shared across space), and therefore an adversarial program corresponds to an update in a low dimensional parameter subspace.

In this paper, we present the first instances of adversarial reprogramming. In Section 2, we discuss related work. In Section 3, we present a training procedure for crafting adversarial programs, which cause a neural network to perform a new task. In Section 4, we experimentally demonstrate adversarial programs that target several convolutional neural networks designed to classify ImageNet data. These adversarial programs alter the network function from ImageNet classification to: counting squares in an image, classifying MNIST digits, and classifying CIFAR-10 images. Next, we examine the susceptibility of trained and untrained networks to adversarial reprogramming. We then demonstrate the possibility of reprograming adversarial tasks with adversarial data that has no resemblance to original data, demonstrating that results from transfer learning do not fully explain adversarial reprogramming. Further, we demonstrate the possibility of concealing adversarial programs and data. Finally, we end in Sections 5 and 6 by discussing and summarizing our results.

## 2 BACKGROUND AND RELATED WORK

### 2.1 ADVERSARIAL EXAMPLES

One definition of adversarial examples is that they are "inputs to machine learning models that an attacker has intentionally designed to cause the model to make a mistake" (Goodfellow et al., 2017). They are often formed by starting with a naturally occuring image and using a gradient-based optimizer to search for a nearby image that causes a mistake (Biggio et al., 2013; Szegedy et al., 2013; Carlini & Wagner, 2017). These attacks can be either *untargeted* (the adversary succeeds when causing any mistake at all) or *targeted* (the adversary succeeds when causing the model to predict a specific incorrect class). Adversarial attacks have been also proposed for other domains like malware detection (Grosse et al., 2017), generative models (Kos et al., 2017), network policies for reinforcement learning tasks (Huang et al., 2017), and network interpretation (Ghorbani et al.,

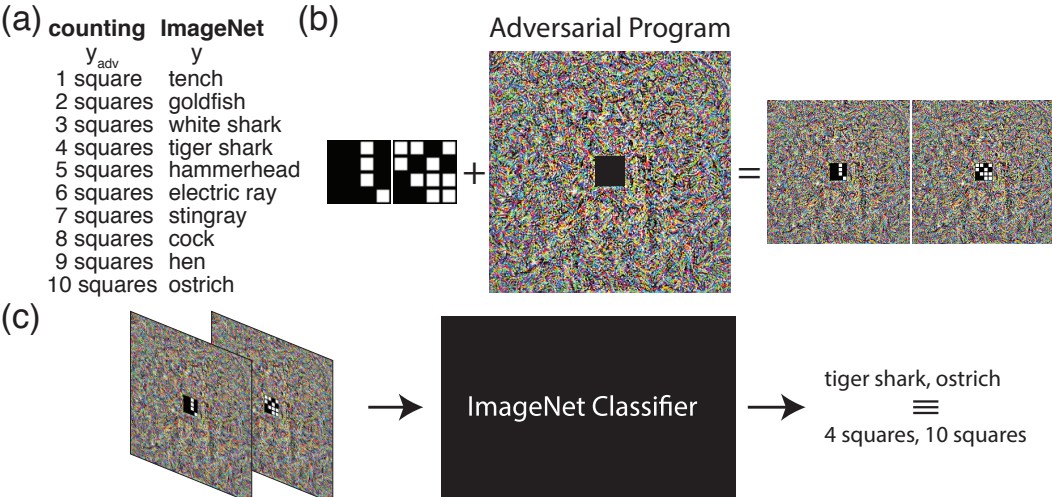

Figure 1: **Illustration of adversarial reprogramming.** (a) Mapping of ImageNet labels to adversarial task labels (squares count in an image). (b) Two examples of images from the adversarial task (left) are embedded at the center of an adversarial program (middle), yielding adversarial images (right). The adversarial program shown repurposes an Inception V3 network to count squares in images. (c) Illustration of inference with adversarial images. The network when presented with adversarial images will predict ImageNet labels that map to the adversarial task labels.

2017). In these domains, the attack remains either untargeted (generally degrading the performance) or targeted (producing a specific output). We extend this line of work by developing reprogramming methods that aim to produce specific *functionality* rather than a specific hardcoded output.

Several authors have observed that the same modification can be applied to many different inputs in order to form adversarial examples (Goodfellow et al., 2014; Moosavi-Dezfooli et al., 2017). For example, Brown et al. (2017) designed an "adversarial patch" that can switch the prediction of many models to one specific class (e.g. toaster) when it is placed physically in their field of view. We continue this line of work by finding a single adversarial program that can be presented with many input images to cause the model to process each image according to the adversarial program.

## 2.2 PARASITIC COMPUTING AND WEIRD MACHINES

Parasitic computing involves forcing a target system to solve a complex computational task it wasn't originally designed to perform, by taking advantage of peculiarities in network communication protocols (Barabasi et al., 2001; Peresini & Kostic, 2013). Weird machines, on the other hand, are a class of computational exploits where carefully crafted inputs can be used to run arbitrary code on a targeted computer (Bratus et al., 2011). Adversarial reprogramming can be seen as a form of parasitic computing, though without the focus on leveraging the communication protocol itself to perform the computation. Similarly, adversarial reprogramming can be seen as an example of neural networks behaving like weird machines, though adversarial reprogramming functions only within the neural network paradigm – we do not gain access to the host computer.

## 2.3 TRANSFER LEARNING

Transfer learning (Raina et al., 2007; Mesnil et al., 2011) and adversarial reprogramming share the goal of repurposing networks to perform a new task. Transfer learning methods use the knowledge obtained from one task as a base to learn how to perform another. Neural networks possess properties that can be useful for many tasks (Yosinski et al., 2014). For example, neural networks when trained on images develop features that resemble Gabor filters in early layers even if they are trained with different datasets or different training objectives such as supervised image classification (Krizhevsky et al., 2012), unsupervised density learning (Lee et al., 2009), or unsupervised learning of sparse representations (Le et al., 2011). Empirical work has demonstrated that it is possible to take a

convolutional neural network trained to perform one task, and simply train a linear SVM classifier to make the network work for other tasks (Razavian et al., 2014; Donahue et al., 2014). However, transfer learning is very different from the adversarial reprogramming task in that it allows model parameters to be changed for the new task. In typical adversarial settings, an attacker is unable to alter the model, and instead must achieve their goals solely through manipulation of the input. Further, one may wish to adversarially reprogram across tasks with very different datasets. This makes the task of adversarial reprogramming more challenging than transfer learning.

## 3  METHODS

In this work, we consider an adversary with access to the parameters of a neural network that is performing a specific task. The objective of the adversary is to reprogram the model to perform a new task by crafting an adversarial program to be included within the network input. Here, the network was originally designed to perform ImageNet classification, but the methods discussed here can be directly extended to other settings.

Our adversarial program is formulated as an additive contribution to network input. Note that unlike most adversarial perturbations, the adversarial program is not specific to a single image. The same adversarial program will be applied to all images. We define the adversarial program as:

$$P = \tanh\left(W \odot M\right) \tag{1}$$

where $W \in \mathbb{R}^{n \times n \times 3}$ is the adversarial program parameters to be learned, $n$ is the ImageNet image width, and $M$ is a masking matrix that is 0 for image locations that corresponds to the adversarial data for the new task, otherwise 1. Note that the mask $M$ is not required – we mask out the central region of the adversarial program purely to improve visualization of the action of the adversarial program. Also, note that we use $\tanh\left(\cdot\right)$ to bound the adversarial perturbation to be in $(-1, 1)$ – the same range as the (rescaled) ImageNet images the target networks are trained to classify.

Let, $\tilde{x} \in \mathbb{R}^{k \times k \times 3}$ be a sample from the dataset to which we wish to apply the adversarial task, where $k < n$. $\tilde{X} \in \mathbb{R}^{n \times n \times 3}$ is the equivalent ImageNet size image with $\tilde{x}$ placed in the proper area, defined by the mask $M$. The corresponding adversarial image is then:

$$X_{adv} = h_f\left(\tilde{x}; W\right) = \tilde{X} + P \tag{2}$$

Let $P(y|X)$ be the probability that an ImageNet classifier gives to ImageNet label $y \in \{1, \ldots, 1000\}$, given an input image $X$. We define a hard-coded mapping function $h_g(y_{adv})$ that maps a label from an adversarial task $y_{adv}$ to a set of ImageNet labels. For example, if an adversarial task has 10 different classes ($y_{adv} \in \{1, \ldots, 10\}$), $h_g\left(\cdot\right)$ may be defined to assign the first 10 classes of ImageNet, any other 10 classes, or multiple ImageNet classes to the adversarial labels. Our adversarial goal is thus to maximize the probability $P(h_g(y_{adv})|X_{adv})$. We set up our optimization problem as

$$\hat{W} = \underset{W}{\operatorname{argmin}} \left(-\log P(h_g(y_{adv})|X_{adv}) + \lambda||W||_F^2\right), \tag{3}$$

where $\lambda$ is the coefficient for a weight norm penalty, to reduce overfitting. We optimize this loss with Adam while exponentially decaying the learning rate. Hyperparameters are given in Appendix A. Note that after the optimization the adversarial program has minimal computation cost for the adversary, as it only requires computing $X_{adv}$ (Equation 2), and mapping the resulting ImageNet label to the correct class. In other words, during inference the adversary needs only store the program and add it to the data, thus leaving the majority of computation to the target network.

One interesting property of adversarial reprogramming is that it must exploit nonlinear behavior of the target model. This is in contrast to traditional adversarial examples, where attack algorithms based on linear approximations of deep neural networks are sufficient to cause a high error rate (Goodfellow et al., 2014). Consider a linear model that receives an input $\tilde{x}$ and a program $\theta$ concatenated into a single vector: $x = [\tilde{x}, \theta]^\top$. Suppose that the weights of the linear model are partitioned into two sets, $v = [v_{\tilde{x}}, v_\theta]^\top$. The output of the model is $v^\top x = v_{\tilde{x}}^\top \tilde{x} + v_\theta^\top \theta$. The adversarial program $\theta$ adapts the effective biases $v_\theta^\top \theta$ but cannot adapt the weights applied to the input $\tilde{x}$. The adversarial program $\theta$ can thus bias the model toward consistently outputting one class or the other but cannot change the way the input is processed. For adversarial reprogramming to work, the model must include nonlinear interactions between $\tilde{x}$ and $\theta$. A nonlinear deep network satisfies this requirement.

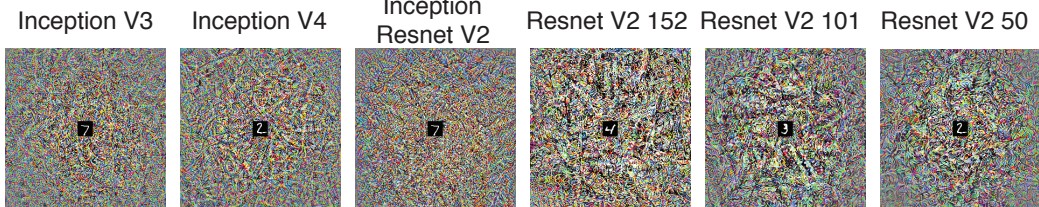

Figure 2: **Examples of adversarial programs for MNIST classification.** Adversarial programs which cause six ImageNet models to function as MNIST classifiers. Each program is shown being applied to an MNIST digit.

## 4 RESULTS

To demonstrate the feasibility of adversarial reprogramming, we conducted experiments on six architectures trained on ImageNet. In each case, we reprogrammed the network to perform three different adversarial tasks: counting squares, MNIST classification, and CIFAR-10 classification. The weights of all trained models were obtained from TensorFlow-Slim, and top-1 ImageNet precisions are shown in Table Supp. 1. We additionally examined whether adversarial training conferred resistance to adversarial reprogramming, and compared the susceptibility of trained networks to random networks. Further, we investigated the possibility of reprogramming the networks when the adversarial data has no resemblance to the original data. Finally, we demonstrated the possibility of concealing the adversarial program and the adversarial data.

### 4.1 COUNTING SQUARES

To illustrate the adversarial reprogramming procedure, we start with a simple adversarial task. That is counting the number of squares in an image. We generated images ($\tilde{x}$) of size $36 \times 36 \times 3$ that include $9 \times 9$ white squares with black frames. Each square could appear in 16 different position in the image, and the number of squares ranged from 1 to 10. The squares were placed randomly on gridpoints (Figure 1b left). We embedded these images in an adversarial program (Figure 1b middle). The resulting images ($X_{adv}$) are of size $299 \times 299 \times 3$ with the $36 \times 36 \times 3$ images of the squares at the center (Figure 1b right). Thus, the adversarial program is simply a frame around the counting task images. We trained one adversarial program per ImageNet model, such that the first 10 ImageNet labels represent the number of squares in each image (Figure 1c). Note that the labels we used from ImageNet have no relation to the labels of the new adversarial task. For example, a 'White Shark' has nothing to do with counting 3 squares in an image, and an 'Ostrich' does not at all resemble 10 squares. We then evaluated the accuracy in the task by sampling 100,000 images and comparing the network prediction to the number of squares in the image.

Despite the dissimilarity of ImageNet labels and adversarial labels, and that the adversarial program is equivalent simply to a first layer bias, the adversarial program masters this counting task for all networks (Table 1). These results demonstrate the vulnerability of neural networks to reprogramming on this simple task using only additive contributions to the input.

### 4.2 MNIST CLASSIFICATION

In this section, we demonstrate adversarial reprogramming on somewhat more complex task of classifying MNIST digits. We measure *test* and train accuracy, so it is impossible for the adversarial program to have simply memorized all training examples. Similar to the counting task, we embedded MNIST digits of size $28 \times 28 \times 3$ inside a frame representing the adversarial program, we assigned the first 10 ImageNet labels to the MNIST digits, and trained an adversarial program for each ImageNet model. Figure 2 shows examples of the adversarial program for each network being applied.

Our results show that ImageNet networks can be successfully reprogramed to function as an MNIST classifier by presenting an additive adversarial program. The adversarial program additionally generalized well from the training to test set, suggesting that the reprogramming does not function purely by memorizing train examples, and is not brittle to small changes in the input.

Table 1: **Neural networks adversarially reprogrammed to perform a variety of tasks.** Table gives accuracy of reprogrammed networks to perform a counting task, MNIST classification task, CIFAR-10 classification task, and Shuffled MNIST classification task.

| Model | Pretrained on ImageNet | | | | | Untrained |
| | Counting | MNIST | | CIFAR-10 | | MNIST |
| | | train | test | train | test | test |
| --- | --- | --- | --- | --- | --- | --- |
| Incep. V3 | 0.9993 | 0.9781 | 0.9753 | 0.7311 | 0.6911 | 0.4539 |
| Incep. V4 | 0.9999 | 0.9638 | 0.9646 | 0.6948 | 0.6683 | 0.1861 |
| Incep. Res. V2 | 0.9994 | 0.9773 | 0.9744 | 0.6985 | 0.6719 | 0.1135 |
| Res. V2 152 | 0.9763 | 0.9478 | 0.9534 | 0.6410 | 0.6210 | 0.1032 |
| Res. V2 101 | 0.9843 | 0.9650 | 0.9664 | 0.6435 | 0.6301 | 0.1756 |
| Res. V2 50 | 0.9966 | 0.9506 | 0.9496 | 0.6 | 0.5858 | 0.9325 |
| Incep. V3 adv. | | 0.9761 | 0.9752 | | | |

## 4.3 CIFAR-10 CLASSIFICATION

Here we implement a more challenging adversarial task. That is, crafting adversarial programs to repurpose ImageNet models to instead classify CIFAR-10 images. Some examples of the resulting adversarial images are given in Figure Supp. 1. Our results show that our adversarial program was able to increase the accuracy on CIFAR-10 from chance to a moderate accuracy (Table 1). This accuracy is near what is expected from typical fully connected networks (Lin et al., 2015) but with minimal computation cost from the adversary side at inference time. One observation is that although adversarial programs trained to classify CIFAR-10 are different from those that classify MNIST or perform the counting task, the programs show some visual similarities, e.g. ResNet architecture adversarial programs seem to possess some low spatial frequency texture (Figure Supp. 2a).

## 4.4 INVESTIGATION OF THE EFFECT OF THE TRAINED MODEL DETAILS AND ORIGINAL DATA

One important question is what is the degree to which susceptibility to adversarial reprogramming depends on the details of the model being attacked. To address this question, we examined attack success on an Inception V3 model that was trained on ImageNet data using adversarial training (Tramèr et al., 2017). Adversarial training augments data with adversarial examples during training, and is one of the most common methods for guarding against adversarial examples. As in Section 4.2, we adversarially reprogrammed this network to classify MNIST digits. Our results (Table 1) indicate that the model trained with adversarial training is still vulnerable to reprogramming, with only a slight reduction in attack success. This finding shows that standard approaches to adversarial defense has little efficacy against adversarial reprogramming. This is likely explained by the differences between adversarial reprogramming and standard adversarial attacks. First, that the goal is to repurpose the network rather than cause it to make a specific mistake, second that the magnitude of adversarial programs can be large, while traditional adversarial attacks are of a small perturbations magnitude, and third adversarial defense methods may be specific to original data and may not generalize to data from the adversarial task.

To further explore dependence on the details of the model, we performed adversarial reprogramming attacks on models with random weights. We used the same experiment set up and MNIST reprogramming task as in Section 4.2 – we simply used the ImageNet models with randomly initialized rather than trained weights. The MNIST classification task was easy for networks pretrained on ImageNet (Table 1). However, for random networks, training was very challenging and generally converged to a much lower accuracy (only ResNet V2 50 could train to a similar accuracy as trained ImageNet models; see Table 1). Moreover, the appearance of the adversarial programs was qualitatively distinct from the adversarial programs obtained with networks pretrained on ImageNet (see Figure Supp. 2b). This finding demonstrates that the original task the neural networks perform is important for adversarial reprogramming. This result may seem surprising, as random networks have rich structure adversarial programs might be expected to take advantage of. For example, theoretical results have shown that wide neural networks become identical to Gaussian processes, where training specific weights in intermediate layers is not necessary to perform tasks (Matthews et al., 2018; Lee et al.,

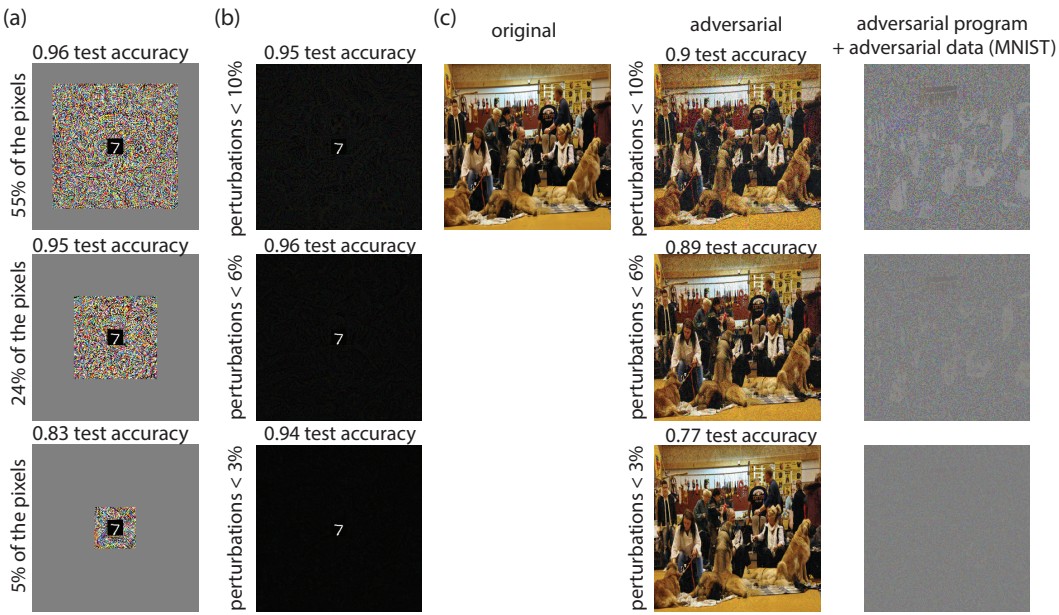

Figure 3: **Adversarial programs may be limited in size or concealed.** In all panels, an Inception V3 model pretrained on ImageNet is reprogrammed to classify MNIST digits. Example images (a) with adversarial programs of different sizes, and (b) with adversarial programs of different perturbation scales. In (c), the adversarial data + program (right) are hidden inside a normal image from ImageNet (left), yielding an adversarial image (center) that is able to reprogram the network to function as an MNIST classifier. The pixels of the adversarial data are shuffled to conceal its structure.

2017). Other work has demonstrated that it is possible to use random networks as generative models for images (Ustyuzhaninov et al., 2016; He et al., 2016), further supporting their potential richness. One explanation may be that randomly initialized networks perform poorly for simple reasons, such as poor scaling of network weights at initialization, whereas the trained weights are better conditioned.

One explanation of adversarial reprogramming that is motivated by transfer learning (Yosinski et al., 2014) is that the network may be relying on some similarities between original and adversarial data. To address this hypothesis, we randomized the pixels on MNIST digits such that any resemblance between the adversarial data (MNIST) and images in the original data (ImageNet) is removed (see Figure Supp. 3). We then attempted to reprogram pretrained ImageNet networks to classify the shuffled MNIST digits. Despite shuffled MNIST not sharing any spatial structure with images, we managed to reprogram the ImageNet network for this task (Table Supp. 7) with almost equal accuracy to standard MNIST (in some cases shuffled MNIST even achieved higher accuracy). We also investigated the possibility of reprogramming shuffled CIFAR-10 images. Our results show that it is possible to reprogram neural networks to classify shuffled CIFAR-10 images (Table Supp. 7). The accuracy for shuffled CIFAR-10 decreased as the convolutional structure of the network is not useful to classify the shuffled images. However, the accuracy was comparable to that expected from fully connected networks (Novak et al., 2019). These results thus suggest that transferring knowledge between the original and adversarial data does not completely explain the susceptibility to adversarial reprogramming. Even more interestingly, these results suggest the possibility of reprogramming across tasks with unrelated datasets and across domains.

## 4.5 CONCEALING ADVERSARIAL PROGRAMS

In our previous experiment, there were no constraints on the size (number of program pixels) or scale (magnitude of perturbations) of the adversarial program. Here, we demonstrate the possibility of limiting the visibility of the adversarial perturbations by limiting the program size, scale, or even concealing the whole adversarial task. In these experiments, we used an Inception V3 model pretrained to classify ImageNet.

In our first experiment, we adversarially reprogrammed the network to classify MNIST digits while limiting the size of the program (Figure 3a). Our results show that adversarial reprogramming is still successful, yet with lower accuracy, even if we use a very small adversarial program. In our next experiment, we made the adversarial program nearly imperceptible by limiting the $L_{\mathrm{inf}}$ norm of the adversarial perturbation to a small percentage of the pixel values. Our results show that adversarial reprogramming is still successful (Figure 3b) even with nearly imperceptible programs.

Further, we tested the possibility of concealing the whole adversarial task by hiding both the adversarial data and program within a normal image from ImageNet. To do this, we shuffled the pixels of the adversarial data (here MNIST), so that the adversarial data structure is hidden. Then, we limited the scale of both the adversarial program and data to a small fraction of the possible pixel values. We added the resulting image to a random image from ImageNet. Formally, we extended our reprogramming method as follows:

$$P_X = \alpha \tanh\left(\mathrm{shuffle}_{ix}(\tilde{X}) + (W \odot \mathrm{shuffle}_{ix}(M))\right) \tag{4}$$

$$X_{adv} = \mathrm{clip}\left(X_{ImageNet} + P_X, \ [-1,1]\right), \tag{5}$$

where $\tilde{X}$, $M$ and $W$ are as described in Section 3, $P_X$ is the adversarial data combined with the adversarial program, $ix$ is the shuffling sequence (same for $M$ and $\forall X$), $\alpha$ is a scalar used to limit the perturbation scale, and $X_{ImageNet}$ is an image chosen randomly from ImageNet, which is the same for all MNIST examples. We then optimized the adversarial program for the network to classify MNIST digits (see Equation 3). The resulting adversarial images are very similar to normal images from ImageNet (see Figure 3c), yet the network is successfully reprogrammed to classify MNIST digits, though with lower accuracy (see Figure 3c). This result demonstrates the possibility of hiding the adversarial task. Here, we used a simple shuffling technique and picked an image from ImageNet to hide the adversarial task, but one could go further and use more complex schemes for hiding the adversarial task and optimize the choice of the image from ImageNet, which may make adversarial reprogramming more effective and harder to detect.

## 5 Discussion

### 5.1 Flexibility of trained neural networks

We found that trained neural networks were more susceptible to adversarial reprogramming than random networks. Further, we found that reprogramming is still successful even when data structure is very different from the structure of the data in the original task. This demonstrates a large flexibility of repurposing trained weights for a new task. Our results suggest that dynamical reuse of neural circuits should be practical in modern artificial neural networks. This holds the promise of enabling machine learning systems which are easier to repurpose, more flexible, and more efficient due to shared compute. Indeed, recent work in machine learning has focused on building large dynamically connected networks with reusable components (Shazeer et al., 2017).

It is unclear whether the reduced performance when targeting random networks, and when reprogramming to perform CIFAR-10 classification, was due to limitations in the expressivity of the adversarial perturbation, or due to the optimization task in Equation 3 being more difficult in these situations. Disentangling limitations in expressivity and trainability will be an interesting future direction.

### 5.2 Adversarial goals beyond the image domain

We demonstrated adversarial reprogramming on classification tasks in the image domain. It is an interesting area for future research whether similar attacks might succeed for audio, video, text, or other domains and tasks. Our finding that trained networks can be reprogrammed to classify shuffled images, which do not retain any of the original spatial structure, suggests that reprogramming across domains is likely possible.

Adversarial reprogramming of recurrent neural networks (RNNs) would be particularly interesting, since RNNs (especially those with attention or memory) can be Turing complete (Neelakantan et al., 2015). An attacker would thus only need to find inputs which induced the RNN to perform a number of simple operations, such as increment counter, decrement counter, and change input attention

location if counter is zero (Minsky, 1961). If adversarial programs can be found for these simple operations, then they could be composed to reprogram the RNN to perform any computational task.

A variety of nefarious ends may be achievable if machine learning systems can be reprogrammed by a specially crafted input. The most direct of these is the theft of computational resources. For instance, an attacker might develop an adversarial program which causes the computer vision classifier in a cloud hosted photos service to solve image captchas and enable creation of spam accounts. If RNNs can be flexibly reprogrammed as mentioned above, this computational theft might extend to more arbitrary tasks. A major danger beyond the computational theft is that an adversary may repurpose computational resources to perform a task which violates the code of ethics of system providers. This is particularly important as ML service providers are invested in protecting the ethical principles and guidelines that governs the use of their services.

## 6 CONCLUSION

In this work, we proposed a new class of adversarial attacks that aim to reprogram neural networks to perform novel adversarial tasks. Our results demonstrate for the first time the possibility of such attacks. They are also illustrative of both surprising flexibility and surprising vulnerability in deep neural networks. Future investigation should address the properties and limitations of adversarial reprogramming, and possible ways to mitigate or defend against it.

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

# Supplemental material

## A    SUPPLEMENTARY TABLES

Table Supp. 1: Top-1 precision of models on ImageNet data

| Model | Accuracy |
|---|---|
| Inception V3 | 0.78 |
| Inception V4 | 0.802 |
| Inception Resnet V2 | 0.804 |
| Resnet V2 152 | 0.778 |
| Resnet V2 101 | 0.77 |
| Resnet V2 50 | 0.756 |
| Inception V3 adv. | 0.776 |

Table Supp. 2: Hyper-parameters for adversarial program training for the square counting adversarial task. For all models, we used the Adam optimizer with its default parameters while decaying the learning rate exponentially during training. We distributed training data across a number of GPUs (each GPU receive 'batch' data samples ). We then performed synchronized updates of the adversarial program parameters.

| ImageNet Model | $\lambda$ | batch | GPUS | learn rate | decay | epochs/decay | steps |
|---|---|---|---|---|---|---|---|
| Inception V3 | 0.01 | 50 | 4 | 0.05 | 0.96 | 2 | 100000 |
| Inception V4 | 0.01 | 50 | 4 | 0.05 | 0.96 | 2 | 100000 |
| Inception Resnet V2 | 0.01 | 50 | 4 | 0.05 | 0.96 | 2 | 100000 |
| Resnet V2 152 | 0.01 | 20 | 4 | 0.05 | 0.96 | 2 | 100000 |
| Resnet V2 101 | 0.01 | 20 | 4 | 0.05 | 0.96 | 2 | 60000 |
| Resnet V2 50 | 0.01 | 20 | 4 | 0.05 | 0.96 | 2 | 100000 |

Table Supp. 3: Hyper-parameters for adversarial program training for MNIST classification adversarial task. For all models, we used the Adam optimizer with its default parameters while decaying the learning rate exponentially during training. We distributed training data across a number of GPUs (each GPU receive 'batch' data samples ). We then performed synchronized updates of the adversarial program parameters. (The Model Inception V3 adv is pretrained on ImageNet data using adversarial training method.

| ImageNet Model | $\lambda$ | batch | GPUS | learn rate | decay | epochs/decay | steps |
|---|---|---|---|---|---|---|---|
| Inception V3 | 0.05 | 100 | 4 | 0.05 | 0.96 | 2 | 60000 |
| Inception V4 | 0.05 | 100 | 4 | 0.05 | 0.96 | 2 | 60000 |
| Inception Resnet V2 | 0.05 | 50 | 8 | 0.05 | 0.96 | 2 | 60000 |
| Resnet V2 152 | 0.05 | 50 | 8 | 0.05 | 0.96 | 2 | 60000 |
| Resnet V2 101 | 0.05 | 50 | 8 | 0.05 | 0.96 | 2 | 60000 |
| Resnet V2 50 | 0.05 | 100 | 4 | 0.05 | 0.96 | 2 | 60000 |
| Inception V3 adv. | 0.01 | 50 | 6 | 0.05 | 0.98 | 4 | 100000 |

Table Supp. 4: Hyper-parameters for adversarial program training for CIFAR-10 classification adversarial task. For all models, we used ADAM optimizer with its default parameters while decaying the learning rate exponentially during training. We distributed training data on number of GPUS (each GPU receive 'batch' data samples ). We then performed synchronized updates of the adversarial program parameters.

| ImageNet Model | $\lambda$ | batch | GPUS | learn rate | decay | epochs/decay | steps |
|---|---|---|---|---|---|---|---|
| Inception V3 | 0.01 | 50 | 6 | 0.05 | 0.99 | 4 | 300000 |
| Inception V4 | 0.01 | 50 | 6 | 0.05 | 0.99 | 4 | 300000 |
| Inception Resnet V2 | 0.01 | 50 | 6 | 0.05 | 0.99 | 4 | 300000 |
| Resnet V2 152 | 0.01 | 30 | 6 | 0.05 | 0.99 | 4 | 300000 |
| Resnet V2 101 | 0.01 | 30 | 6 | 0.05 | 0.99 | 4 | 300000 |
| Resnet V2 50 | 0.01 | 30 | 6 | 0.05 | 0.99 | 4 | 300000 |

Table Supp. 5: Hyper-parameters for adversarial program training for MNIST classification adversarial task. For all models, we used the Adam optimizer with its default parameters while decaying the learning rate exponentially during training. We distributed training data across a number of GPUs (each GPU receive 'batch' data samples ). We then performed synchronized updates of the adversarial program parameters.

| Random Model | $\lambda$ | batch | GPUS | learn rate | decay | epochs/decay | steps |
|---|---|---|---|---|---|---|---|
| Inception V3 | 0.01 | 50 | 4 | 0.05 | 0.96 | 2 | 100000 |
| Inception V4 | 0.01 | 50 | 4 | 0.05 | 0.96 | 2 | 100000 |
| Inception Resnet V2 | 0.01 | 50 | 4 | 0.05 | 0.96 | 2 | 60000 |
| Resnet V2 152 | 0.01 | 20 | 4 | 0.05 | 0.96 | 2 | 60000 |
| Resnet V2 101 | 0.01 | 20 | 4 | 0.05 | 0.96 | 2 | 60000 |
| Resnet V2 50 | 0.01 | 50 | 4 | 0.05 | 0.96 | 2 | 60000 |

Table Supp. 6: Hyper-parameters for adversarial program training for Shuffled CIFAR-10 classification task. For all models, we used 1 GPU and a batch size of 100. We decayed the learning rate with a cosine schedule over 200 epochs. We performed hyperparameters tuning and picked the best parameters based on a held-out validation set. The best parameters are shown below.

| ImageNet Model | $\lambda$ | optimizer | learn rate |
|---|---|---|---|
| Inception V3 | 0 | Momentum | 0.1 |
| Inception V4 | 0.1 | Momentum | 0.001 |
| Inception Resnet V2 | 0 | ADAM | 0.001 |
| Resnet V2 152 | 0.1 | Momentum | 0.1 |
| Resnet V2 101 | 0.01 | Momentum | 0.1 |
| Resnet V2 50 | 0.01 | Momentum | 0.1 |

Table Supp. 7: **Neural networks adversarially reprogrammed to perform classification tasks with shuffled pixels.** Table gives accuracy of reprogrammed networks to perform classification using Shuffled MNIST and Shuffled CIFAR-10. Shuffling is performed across pixels. The optimization parameters for shuffled MNIST are the same as in Table Supp. 3. The optimization parameters for Shuffled-CIFAR-10 are shown in Table Supp. 6

| Model | Pretrained on ImageNet | |
| --- | --- | --- |
| | Shuffled MNIST | Shuffled CIFAR-10 |
| Incep. V3 | 0.9709 | 0.5578 |
| Incep. V4 | 0.9715 | 0.5618 |
| Incep. Res. V2 | 0.9683 | 0.5507 |
| Res. V2 152 | 0.9691 | 0.5624 |
| Res. V2 101 | 0.9678 | 0.5612 |
| Res. V2 50 | 0.9717 | 0.5614 |

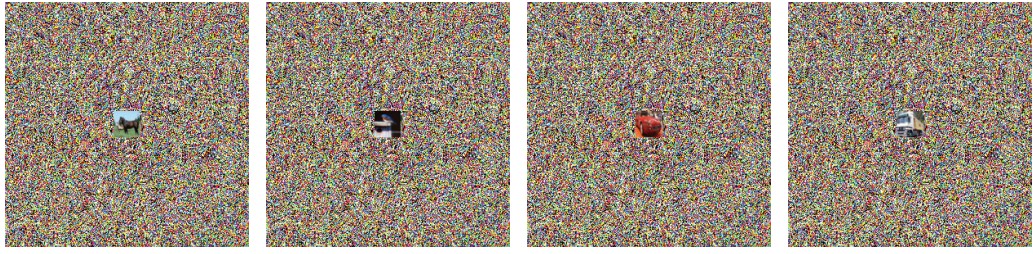

Figure Supp. 1: **Examples of adversarial images for CIFAR-10 classification.** An adversarial program repurposing an Inception V3 model to classify CIFAR-10 images, applied to four CIFAR-10 images.

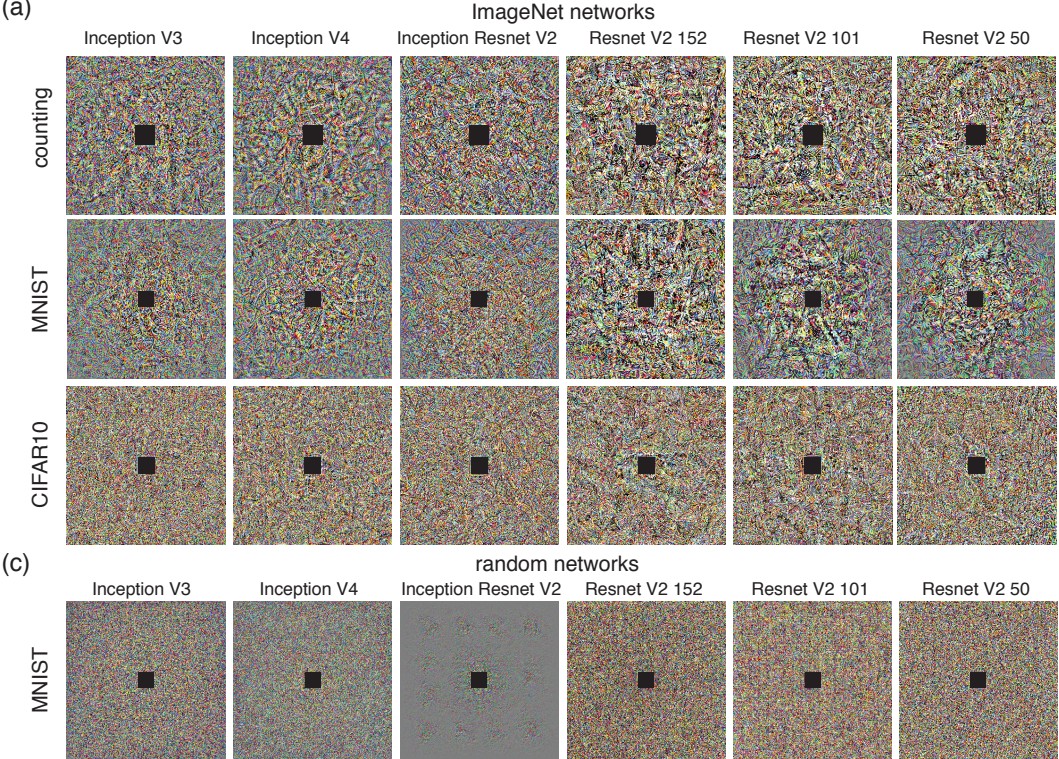

Figure Supp. 2: **Adversarial programs exhibit qualitative similarities and differences across both network and task.** (a) Top: adversarial programs targeted to repurpose networks pre-trained on ImageNet to count squares in images. Middle: adversarial programs targeted to repurpose networks pre-trained on ImageNet to function as MNIST classifiers. Bottom: adversarial programs to cause the same networks to function as CIFAR-10 classifiers. (b) Adversarial programs targeted to repurpose networks with randomly initialized parameters to function as MNIST classifiers.

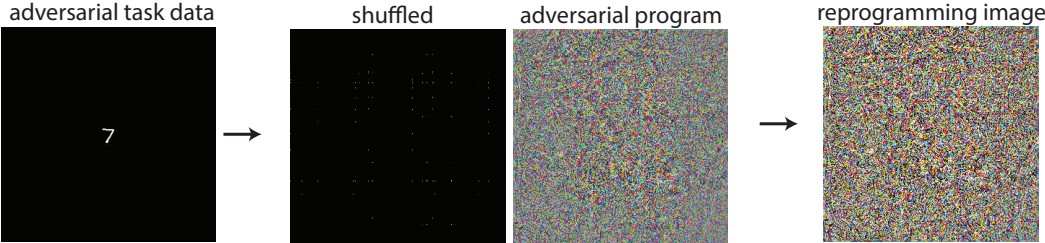

Figure Supp. 3: **Neural networks are susceptible to adversarial reprogramming even in cases when adversarial data and original task data are unrelated.** The pixels in MNIST digits are shuffled. So, that the resulting image has no resemblance to any image. Then, the shuffled image is combined with the adversarial program to create a reprogramming image. This image successfully reprogram Inception V3 model to classify the shuffled digits, despite that the adversarial data (i.e., shuffled MNIST digits) being unrelated to the original data (i.e., ImageNet).

