# OpenReview forum: "Adversarial Reprogramming of Neural Networks"
_ICLR.cc/2019/Conference_

### Official Review · AnonReviewer1 · 2018-11-01
**Review for "Adversarial Reprogramming of Neural Networks" -- A Good paper with an interesting novel approach to adversarial attacks**

**Rating:** 8
**Confidence:** 4

**Review:**

Summary:
The authors present a novel adversarial attack scheme where a neural net is repurposed or "reprogrammed" to accomplish a different task than it the one it was originally trained on. This reprogramming from task1 to task2  is done through a given image from task2 additively enhanced with an adversarial program which is trained given the knowledge of the models parameters. A mapping from the repurposed output from task1 to relevant output for taks2 is also necessary (h_g function).

Review:
This approach seems quite novel as it enables the repurposing of ImageNet classifiers to be used for counting dots in images, MNIST and CIFAR10 classifications. This new type of "adversarial attack" by repurposing a model shows surprising efficacy at allowing an attacked models to change its task at hand. Some tasks being more difficult (CIFAR10) than MNIST or counting dots.

The paper is well-written and explains clearly the proposed technique. The proposed technique is simple in its formulation.
The assumption it is based on (access to model parameters) is acceptable for the sake of proof of concept.
Overall it is an interesting paper to read and seems of significance for the community working on adversarial attacks.

Few comments/questions come to mind though:
- The adversarial images are quite different from a common image as they embed the program around the new task images. This makes the technique itself quite susceptible to detection (just look at the statistics of the input images).
- How do you handle front end processing? Usually for ImageNet classification, a system will (for instance) resize its input to 256x256, center crop to 224x224 and renormalize the RGB features to match the statistics from the training data. It looks like the images generated are passed as inputs to the system. Do you assume that the front-end steps are not applied or do you assume it is (by including them in the network while training your program W).  My assumption is that you include those steps in the training network for W.
- The size of the program is disproportionately big compare to the task2 embedded image. This begs the question: what happens when you limit the size of the program to a smaller percentage of the whole image? When do you see a break in the reprogramming? Do you need that much extra programming W in your adversarial images?
- As the adversarial images seem to be quite easy to detect, would it be easy to integrate it into some task1 images? The equation (2) gives X_{adv} = \tilda{X} + P, could you use X_{adv} + w * X_{task1}, basically finding a way to hide the program and task2 image within a task1 image. This seems difficult, but have you thought of such approach?

Overall this is a paper that is a pleasant read and should be considered for publication.

Post Rebuttal: The draft paper improves on the original paper and demonstrates possible concealment of the program. I adjusted my rating upward to 8.

---

> ### Author Response · Authors · 2018-11-20
> **Response to reviewer 1**
>
> Thank you very much for your feedback.
>
> To address your comment about the adversarial program being quite different from a common image, we conducted the suggested experiment and demonstrated that the whole adversarial program and data could be hidden within normal images (for details, see the response to reviewer 2 and new Section 4.5). We believe this also addresses your last comment about hiding the program.
>
> For the front end processing, we apply our adversarial program after preprocessing images. However, one can equally apply the program before these steps and incorporate the preprocessing into the optimization objective.
>
> Regarding your comment about the size of the program, we conducted new experiments where we limited the size of the program to 55%, 24%, and 5% of the image size (Section 4.5). Our results show that the networks are still susceptible to adversarial reprogramming, yet accuracy decreases as we limit the program size. On the broader point of complexity of the program and successful of reprogramming, one may increase programming capacity by using more complex adversarial programs rather than simple perturbations.
>
> We believe the new manuscript largely addresses your comments; please see our updated manuscript and consider updating your score as appropriate. Thank you very much for your comments and suggestions.

---

> > ### Comment · AnonReviewer1 · 2018-12-03
> > **Response to the Authors**
> >
> > Thanks for addressing my comments and adding section 4.5 that demonstrate the impact of the size of the program and the possibility of concealment in the image. I think this makes it a better paper and I will adjust my rating accordingly

---

### Official Review · AnonReviewer2 · 2018-11-03
**"adversarial reprogramming" should be better cast as a trainable input perturbation on a fixed network for multi-task learning; the contribution is unclear and the "adversarial" setting is not well motivated**

**Rating:** 6
**Confidence:** 5

**Review:**

This paper proposed "adversarial reprogramming" of well-trained and fixed neural networks, which can be viewed as learning a trainable input perturbation on a fixed network for multi-tasking by using a different dataset (e.g., MNIST) from the original dataset (ImageNet) as input. Domain mapping functions (h_g and h_f) are required if the data have different dimensions. The key factor to enable adversarial reprogramming of a fixed network to perform a different task is by training the additive adversarial program as defined in (1). Experimental results show that 7 different ImageNet models (adversarially trained or not) can be reprogrammed for performing counting tasks, and MNIST and CIFAR-10 classifications. The authors also show that adversarial reprogramming is less effective on untrained networks.

Although the idea of this paper is interesting,  the contribution is unclear and the "adversarial" setting is not well motivated. The detailed comments are as follows.

1. Unclear contribution - As mentioned in this paper, the main difference between "adversarial reprogramming" and transfer learning or multi-task learning is the fact that the network to be reprogrammed is fixed during reprogramming and was trained on a single task that is independent of the targeted task. However, the reprogramming results are not surprising given the fact that multi-task learning can be achieved on the same network. Given the fact that the perturbed input data (e.g., MNIST) is different from the original input data (ImageNet), what adversarial reprogramming demonstrates is actually a simple way of learning a new task via input perturbation to an unseen dataset at training time. However, transfer learning can be done in a similar way by simply fine-tuning the last (few) layers of a well-trained network. So the number of parameters required to be modified in order to "reprogram" a network is already known to be quite small via fine-tuning, which may even be less than the dimension of the adversarial program. In addition, given that the input of ImageNet model is high-dimensional and ImageNet images are likely to lie on a low dimensional manifold (but they are very different from hand-written digits or CIFAR images), the capability of reprogramming using deep models under this setting is expected and thus the contribution is unclear.

2. The "adversarial" setting is vague - I am very confused about why the experimental settings should be considered "adversarial", given the fact that ImageNet images and the three sets of adversarially perturbed images are quite different. What the experiments show is that a well-trained classifier has a large enough capacity to perform other tasks by simply training a perturbation on a different (out-of-distribution) dataset as inputs. It would make more sense to call this method "adversarial" if it can be used on ImageNet images to secretly implement some programmed tasks, while on the surface they are seemingly simply performing a typical classification task.

3. Limited novelty - How is adversarial program different from additional perturbation? Let alone the mapping function M in eqn (3), the adversarial program is nothing but a constrained perturbation (ranging from [-1,1] in each dimension). The optimization formulation in (3) can be seen as a  Carlini-Wager L2 attack with a simplified attack loss + L2 distortion regularization. Therefore, the proposed method has limited technical contribution and novelty.

In summary, this paper has some interesting ideas, but the current presentation lacks clear motivation, and its technical contribution and implications need to be better highlighted.  The authors are suggested to better motivate this paper from the angle of studying the learning capacity of input perturbation induced multi-tasking learning of a well-trained and fixed neural network model, and compare the pros and cons with transfer learning based on fine-tuning and joint multi-task learning / meta-learning on the same network architecture. Based on my own reading, I truly feel that advocating  "adversarial" reprogramming does not add any value to this work, as its use for an adversary is not properly motivated (e.g., visual imperceptibility) and its training has no adversarial nature (e.g., GAN training). Titles like "(Out-of-domain) Input perturbation induced reprogramming of neural networks" should better justify the contents and experiments presented in this work. Lastly, the authors need to specify how equation (3) is different from the formulation of finding adversarial perturbations in existing literature. Otherwise,  the novelty of "adversarial program" is quite limited.

----
Post-rebuttal review:

I appreciate the authors' efforts in including the new experiments in Sections 4.4 and 4.5. In my opinion, these new results and the discussion in Section 5.2 add great values to this work and make the contributions of this paper substantially clear. I've increased my rating to 6.

---

> ### Author Response · Authors · 2018-11-20
> **Response to reviewer 2**
>
> Thank you for your comments. We have conducted new experiments and made changes to the paper to address your comments (please see the revised paper). We detail these changes below:
>
> As described in the response to Reviewer 3, we have articulated the distinction from transfer learning and included additional supporting experiments (see detailed response to reviewer 3 above). In summary, adversarial reprogramming should be viewed as being more analogous to adversarial examples than to transfer learning. In further experiments, we shuffled data to ensure that it did not have any resemblance to the images used to train these networks. In this shuffled-input-pixels context, transfer learning is meaningless. Our results demonstrate that adversarial reprogramming is still possible, which suggests that classic transfer learning does not explain adversarial reprogramming (Section 4.4).
>
>
> In response to the comment about novelty, we acknowledge and agree that adversarial reprogramming adds perturbation to images, similar to a large body of adversarial methods (also distinct from transfer learning). We cite these related works throughout the paper, and have also added a citation to the Carlini-Wager L2 attack paper for completeness. The novelty of this work lies in the fact that this is the first paper to anticipate and demonstrate the feasibility of a new class of adversarial goals aimed at repurposing networks to run a function desired by an attacker. We show that, even with simple optimization methods, one can achieve this adversarial goal. The simplicity of these methods should highlight the real security vulnerability demonstrated, as one can perform this attack with ease.  We believe this simplicity to be central to the veracity of our claims, and do not agree that it detracts from the novelty of this work.
>
> To address your point on the adversarial setting not being clear, we have expanded the discussion section and performed new experiments to clarify this. In the revised Discussion, we explain that the simplest adversarial goal could be the abuse and theft of computing resources. A more malicious goal would be using ML services in a way that violates the terms of service or ethical restrictions stipulated by the service provider. With many companies now offering accessible ML services, our work brings attention to the fact that training a model on one task is not a guarantee that it will be used only for this task; an adversary could reprogram the model even through simple input interactions.
>
> Further, we conducted new experiments and added a new results section where we applied constraints to the adversarial perturbations, restricting the number of pixels they could span or the scale of the perturbations (Section 4.5). Even with constrained adversarial perturbations, networks are still susceptible to adversarial reprogramming. As further evidence, we show that the whole adversarial task (ie adversarial data and program) can be hidden in a  normal image from ImageNet.
> We believe our improvements to the manuscript and additional experiments have addressed your concerns. Thank you again for your comments, and we hope you will raise your score as a result.

---

> ### Public Comment · ~Nicholas_Carlini1 · 2018-11-20
> **On the Lagrangian relaxation**
>
> I just saw this review and thought I'd comment briefly on the question about how this is related to the L2 attack we proposed in (Carlini & Wagner 2017).
>
> We weren't the first ones to come up with the reformulation loss + lambda*distortion: this is a common trick in optimization, and is provably correct when the problem is sufficiently constrained, see https://en.wikipedia.org/wiki/Lagrangian_relaxation (even in adversarial examples, we weren't first: Szegedy et al. 2013 does it too).
>
> [I don't intend this comment to reflect positively or negatively on any aspect of the paper. I just want to clarify that this idea shouldn't be given to us.]
>
> edit: missing should*n't* be given to us.

---

### Official Review · AnonReviewer3 · 2018-11-04
**Adversarial Reprogramming**

**Rating:** 4
**Confidence:** 3

**Review:**

This paper extends the idea of 'adversarial attacks' in supervised learning of NNs, to a full repurposing of the solution of a trained net.

The note of the authors regarding 'Transfer learning' is making sense even to the extend that I fail to see how the proposed study differs from the setting of Transfer learning. The comment of 'parameters' does not make much sense in a semi-parametric approach as studied. The difference might be significant, but I leave it up to the authors to formulate a convincing argument.

---

> ### Author Response · Authors · 2018-11-20
> **Response to reviewer 3**
>
> Thank you very much for your comments; to address them we have made modifications to the manuscript, including conducting new experiments. We detail these below:
>
> Adversarial reprogramming differs from transfer learning primarily because it focuses on  finding a transformation of model  input such that the transformed input results in changes to the model unction. In contrast, transfer learning is concerned with changing the network parameters to perform a new task. In this manuscript, we transformed the input by adding perturbations designed to repurpose the network function. Thus, our adversarial reprogramming scheme should be viewed in the same way adversarial examples are viewed -- they are perturbations of the input, not tuning of model parameters. We have clarified these points in the new version of the paper (please see the revision).
>
> Further, we conducted new experiments where we shuffled the adversarial data (i.e., MNIST) to remove any resemblance of the adversarial data and the original data (i.e., ImageNet). Our new results show that networks are still susceptible to adversarial reprogramming even when the original data and adversarial data do not share any spatial structure. This demonstrates that transferring learned features of the network from original to new adversarial data does not explain the susceptibility of neural networks to adversarial reprogramming (see the new Section 4.4). Section 4.5 was also added to demonstrate the possibility of concealing adversarial programs.
>
> Thank you again for your feedback. We believe the updated paper convincingly addresses the concerns you raised, and we hope you will raise your score as a result.

---

### Public Comment · (anonymous) · 2018-10-02
**Interesting work! But is this kind of attack really applicable in real world?**

This paper shows a new possibility of attacking neural network models. However, there is a real world concern:

The trained adversarial 'program' seems to take up >95% of the input image size. How would the success rates change as the area of the 'program' changes? The input fed into neural network can be divided into the adversarial 'program' and the de facto input. If the 'program' is too big, the input would be limited. The information allowed in the de facto input is thus limited.

Specifically, consider the following situation where I want to perform a fine-grained classification using this method. On the one hand, if I constrain the size/area of the 'program', success rate may drop significantly; on the other hand, if I constrain the size/area of the image to be classified, it may also fail as details of the image is lost.

The balance between the functionality of the 'program' and the amount of information allowed in the input image doesn't seem a piece of cake.

---

> ### Author Response · Authors · 2018-10-02
> **Response to: Interesting work! But is this kind of attack really applicable in real world?**
>
> Thank you for your comment. We believe that the success rate of the reprogramming may drop as the size of program decreases. However, we believe that one could recover the success rate by increasing the complexity of the the function computing the adversarial program h_f (refer to our introduction). In our experiments, we intentionally used an additive function h_f to demonstrate the severity of the problem as with even this simple transformation adversarial reprogramming is possible. However, as we mention in our introduction the idea of adversarial programming is more general, and more complex functions may not require a large number of adversarial parameters or even that the size of the image to be processed be smaller than the original input size. More generally, we believe it would be a very interesting future research direction to study the effect of both the original model capacity and the complexity of the adversarial functions on the success rate of adversarial reprogramming.

---

> ### Author Response · Authors · 2018-12-15
> **New Section 4.5 and Figure 3**
>
> Note, in the new Section 4.5 and Figure 3 we show that adversarial programs may be limited to a small fraction of the pixels or even made largely imperceptible by restricting magnitude.

---

### Public Comment · (anonymous) · 2018-12-12
**Questions**

Interesting paper! I have the following two questions:

- An MNIST neural network achieving 97% accuracy has on the order of tens of thousands of trainable parameters, and similarly for a network attaining 60-70% on CIFAR. The adversarial programs, by contrast seem to have 299*299*3, or about 300k parameters. As such, is it fair to think of reprogramming as  just training a neural network (albeit with a weird architecture ) and then visualizing the weights? (In this case the weights are where the image used to be, and the inputs are just the old weights).

- In the counting squares task, there are \sum_{k=1}^10 (16 choose k) < 59000  images that could possibly occur—How many examples was the program trained from in the paper (I could not find a number)? Either way, isn't evaluating on 100k images redundant, since we are guaranteed to se 41,000 repeated images?

---

> ### Author Response · Authors · 2018-12-15
> **Response to "Questions"**
>
> Yes, the adversarial program can be thought of as parameters for a particularly bizarre neural net architecture. We note in the last paragraph of Section 3, and in the last paragraph of Section 4.1, that the adversarial program can be seen as equivalent to a particular choice of neural network input layer biases (for a spatially varying bias). Note though that it was not a priori clear that a neural network whose only trainable parameters are input biases can perform well or be trained effectively. Additionally, in the new Section 4.5 and Figure 3 we show that adversarial programs may be limited to a small fraction of the pixels or even made largely imperceptible by restricting magnitude, which further reduces their similarity to standard NN parameterizations.
>
> For your question on the counting task, this task was meant to be an illustration of the idea, and we evaluate on 100k images because we generate the patterns randomly. We agree that the network may solve the task largely by memorizing the patterns. However, this is not the case in all the other tasks. We mention this at the beginning of Section 4.2 “We measure test and train accuracy, so it is impossible for the adversarial program to have simply memorized all training examples.”

---

### Meta-Review · Area_Chair1 · 2018-12-15

**Confidence:** 4
**Recommendation:** Accept (Poster)

**Metareview:**

Reviewers mostly recommended to accept after engaging with the authors. I have decided to reduce the weight of AnonReviewer3 because of the short review. Please take reviewers' comments into consideration to improve your submission for the camera ready.